

# Recent advances in biochar application for water and wastewater treatment: a review

Xiaoqing Wang, Zizhang Guo, Zhen Hu and Jian Zhang

Shandong Key Laboratory of Water Pollution Control and Resource Reuse, School of Environmental Science & Engineering, Shandong University, Qingdao, P.R.China

## ABSTRACT

In the past decade, researchers have carried out a massive amount of research on the application of biochar for contaminants removal from aqueous solutions. As an emerging sorbent with great potential, biochar has shown significant advantages such as the broad sources of feedstocks, easy preparation process, and favorable surface and structural properties. This review provides an overview of recent advances in biochar application in water and wastewater treatment, including a brief discussion of the involved sorption mechanisms of contaminants removal, as well as the biochar modification methods. Furthermore, environmental concerns of biochar that need to be paid attention to and future research directions are put forward to promote the further application of biochar in practical water and wastewater treatment.

## INTRODUCTION

Biochar with rich carbon content is a thermal decomposition product derived from biomass under a condition that lacks oxygen (*Sohi, 2012*). These innovations about converting organic matters into valuable materials such as biochar and the subsequent applications have drawn the attention of relevant fields. Initial studies have focused on the ability of biochars as soil amendments to sorb inorganic nutrients and improve the soil quality or promote other environmental services (*Sanroman et al., 2017*). Numerous researches have shown the interests of biochar in improving soil properties and increasing crop yield (*Windeatt et al., 2014*; *Agegnehu, Srivastava & Bird, 2017*; *Awad et al., 2017*; *ÖZ, 2018*; *Yu et al., 2019*), which ultimately contributes to soil carbon sequestration and reduction of greenhouse gases (*Windeatt et al., 2014*). In recent years, progress in the production of various biochars has improved their performance and expanded their application in multidisciplinary fields. Researches on biochar are being carried out in more and more countries, with broad and diverse purposes depending on the feedstocks, production and modification methods, and the local economy and environment (*Tan et al., 2015*).

Water and wastewater treatment is one of the emerging subsets of biochar application. Due to the properties of large surface area and pore volume, rich organic carbon content and

Corresponding author
Jian Zhang, zhangjian00@sdu.edu.cn

mineral components, abundant and diverse functional groups, biochar displays prominent sorption ability for both inorganic and organic contaminants in aqueous solutions (*Ahmad et al., 2014*). Traditional techniques for contaminants removal from the aqueous phase, for example, ion exchange, membrane separation, chemical precipitation, and sorption using activated carbon, have disadvantages such as high cost and inevitable generation of a large number of chemical residues with no economic value (*Oliveira et al., 2017*). In contrast, biochar can be produced from green wastes, mainly agricultural biomass and solid wastes such as woodchips, straws, shells, bagasse, and manure (*Ahmad et al., 2014*; *Nanda et al., 2016*; *Thornley, Upham & Tomei, 2009*). The resources of feedstocks are among the wealthiest renewable resources in the ecosystem (*Yao et al., 2012*; *Xu et al., 2013*), providing more options to produce such renewable sorbents, which benefits the low-income communities to some extent. *Moreira, Noya & Feijoo (2017)* compared the global environmental impacts between the production process of biochar and activated carbon. The main findings encouraged biochar to be an environmentally friendly alternative to activated carbon, mainly reflected by net mitigation of carbon emissions in the biochar production process.

Biochar is a by-product of thermochemical transformation such as pyrolysis, hydrothermal carbonization, gasification, and torrefaction (*Meyer, Glaser & Quicker, 2011*). Reports have shown that the physicochemical properties of biochar such as surface area, porosity, acid–base behavior, surface functional groups, and element composition depend on pyrolysis temperatures and feedstock types (*Ahmad et al., 2014*; *Uchimiya, Ohno & He, 2013*; *Nachenius et al., 2013*; *Manariotis, Fotopoulou & Karapanagioti, 2015*), making vital implications on its efficiency and suitability in removal of target contaminants (*Oliveira et al., 2017*), including a series of organic contaminants (e.g., dyes, phenols, polycyclic aromatic hydrocarbons (PAHs), pesticides, antibiotics) and inorganic contaminants (e.g., heavy metals, nitrate ($NO_3^-$), ammonium ion ($NH_4^+$), phosphate ($PO_4^{3-}$), fluoride ($F^-$)) from wastewater (detailed discussion is given in the section "Biochar application in water and wastewater treatment").

With increased application of biochar being carried out in the water and wastewater treatment, this paper reviews recent advances in the biochar application, including a brief discussion of the involved mechanisms in the removal of specific organic and inorganic contaminants. Moreover, this review briefly covers the modification methods of biochar based on different emphases and explains how the modification alters the properties of biochar, as well as the removal efficiency. Furthermore, remained environmental concerns and future research directions are highlighted, with possible solutions put forward.

## SURVEY METHODOLOGY

The literature reviewed in this paper was obtained on databases of ScienceDirect, Web of Science, Google Scholar, and the Chinese journal database CNKI. The keywords used to search for literature on the databases are as follows: biochar, cellulose, lignin, pyrolysis, and carbonization associated with the feedstocks and biochar production methods; industrial, agricultural, pharmaceutical, heavy metals, dyes, pesticides, antibiotics, and persistent

contaminants reflecting biochar application; electrostatic interaction, precipitation, complexation, hydrophobic effect, and chemical bonds referred to sorption mechanisms; porosity, surface area, functional groups, magnetization, and biochar-based composites related to modification methods. Besides, literature research was specially conducted within the papers on "Special Issue on Biochar: Production, Characterization and Applications - Beyond Soil Applications" published on "Bioresource Technology," and papers published on "Journal of Chemical Technology and Biotechnology," which were presented in the 2017 European Geosciences Union session "Novel Sorbents for Environmental Remediation" (*Sanroman et al., 2017*; *Manariotis, Karapanagioti & Werner, 2017*).

## SORPTION MECHANISMS

The sorption ability of biochar for contaminant removal has been well documented. However, there are lacking studies on corresponding sorption mechanisms for target contaminants, which have fundamental meanings for improving the removal efficiency. Sorption mechanisms vary according to the properties of both contaminants and biochar. Here, the dominant mechanisms in the removal of heavy metals and organic contaminants are illustrated in Fig. 1.

### Heavy metals

Heavy metals in the water environment mostly come from anthropogenic activities such as smelting, mining, and electronic manufacturing effluents (*Li et al., 2017*). Biochar has been suggested to be used for heavy metals removal from contaminated water. Removal mechanisms vary depending on the valence state of the target metal at different solution pH (*Li et al., 2017*). Four mechanisms dominating heavy metals removal from water by biochar are proposed as follows (*Qian et al., 2015*; *Tan et al., 2015*; *Li et al., 2017*): (i) electrostatic attraction between heavy metals and biochar surface; (ii) ion exchange between heavy metals and alkali or alkaline earth metals or protons on biochar surface; (iii) complexation with $\pi$ electron-rich domain or surface functional groups; (iv) co-precipitation to form insoluble compounds. Here, specific examples are used to explain each mechanism.

Solution pH could strongly influence the surface charge of biochar. $pH_{PZC}$ is the solution pH at which the net charge of the biochar surface is zero. Biochar is positively charged at solution pH < $pH_{PZC}$ and binds metal anions such as $HAsO_4^{2-}$ and $HCrO_4^{-}$. On the contrary, biochar is negatively charged at solution pH > $pH_{PZC}$ and binds metal cations such as $Hg^{2+}$, $Pb^{2+}$, and $Cd^{2+}$ (*Li et al., 2017*). These processes are the electrostatic attraction. For instance, *Wang et al. (2015)* applied pinewood biochar pyrolyzed at 600 °C ($pH_{PZC}$ >7) to sorb As(V) from water at pH 7, with a maximum sorption capacity of 0.3 mg g$^{-1}$. As(V) mainly exists in the form of $HAsO_4^{2-}$ at pH 7. The biochar surface is positively charged since the solution pH < $pH_{PZC}$. In that case, $HAsO_4^{2-}$ interacts with the protonated functional groups on biochar surface by electrostatic attraction.

Biochar pyrolyzed from biomass has plenty of exchangeable cations on the surface, such as some alkali or alkaline earth metals (Na, K, Mg, Ca) that can be replaced by heavy metal ions during the sorption. *Lu et al. (2012)* studied mechanisms for Pb sorption by sludge-derived biochar. They found a certain amount of $Na^+$, $K^+$, $Mg^{2+}$, and $Ca^{2+}$ released
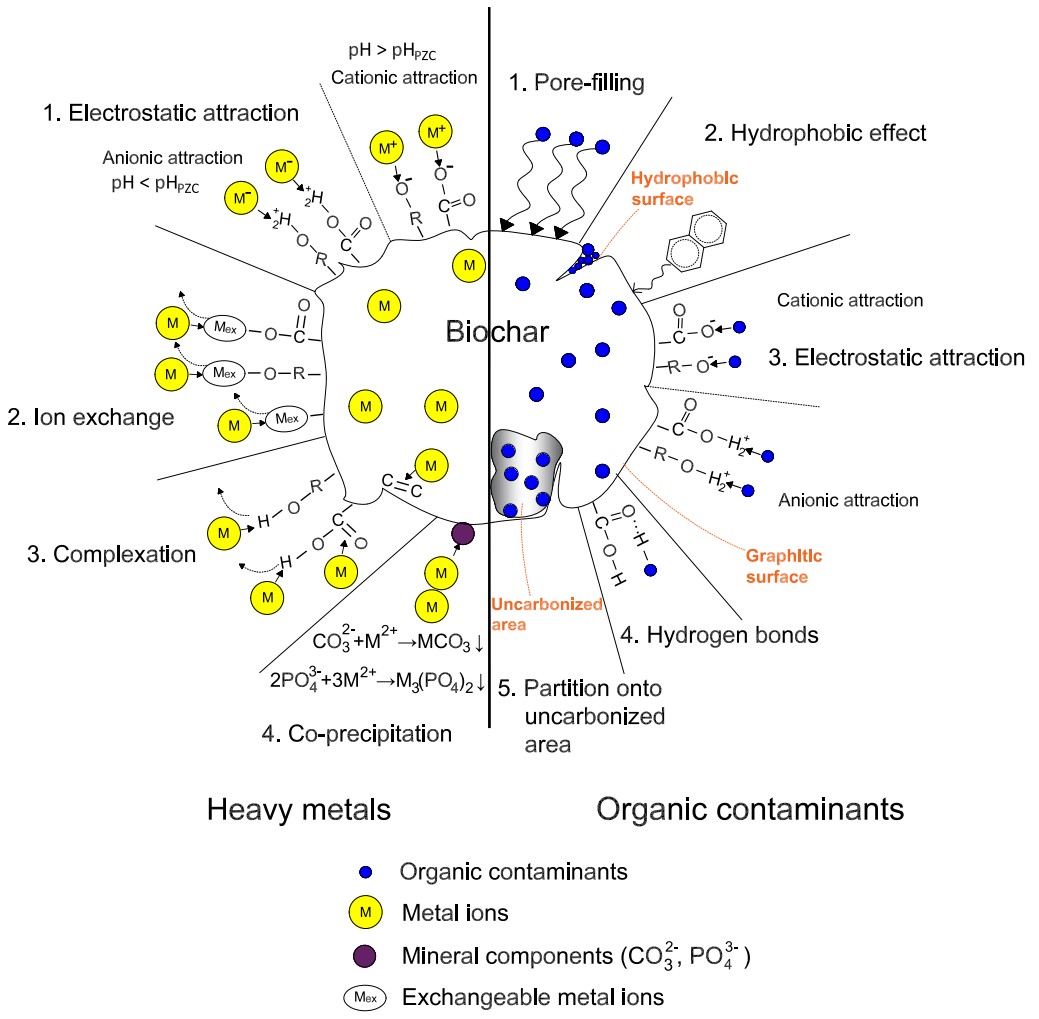

**Figure 1  Sorption mechanisms of heavy metals and organic contaminants on biochar.**

from the biochar, probably as a result of metal exchanges with $Pb^{2+}$. *Zhang et al. (2015)* studied mechanisms for Cd sorption and showed that there was almost an equal amount of sorbed Cd and total released cations (Na, K, Mg, Ca) from the biochar, indicating the cation exchange as a leading role in Cd sorption.

*Xu et al. (2016)* compared different complexation mechanisms of Hg sorption on bagasse and hickory chips biochar. X-ray photoelectric spectroscopy (XPS) showed that the formation of $(-COO)_2Hg$ and $(-O)_2Hg$ attributed mostly to Hg sorption on bagasse biochar. The sorption capacity decreased by 18% and 38% when using methanol to block -COOH and -OH functional groups. Nevertheless, the blocking did not affect Hg sorption on hickory chips biochar since the formation of Hg-$\pi$ bindings between Hg and $\pi$ electrons of C=O and C=C dominated the sorption. *Pan, Jiang & Xu (2013)* investigated Cr(III) sorption on several crop straws biochars. The order of their sorption capacity was

in accordance with the abundance of oxygen-containing functional groups, suggesting the importance of Cr(III) complexation with functional groups.

Mineral components in biochar are also crucial in the removal process, which acts as other sorption sites and makes contributions to heavy metals sorption by precipitation (*Xu et al., 2013*). For example, precipitation was implied to be the dominant mechanism for Cd removal on dairy manure biochar owing to its relatively high soluble carbonate and phosphate content (*Xu et al., 2013*). With the temperature increasing from 200 to 350 °C, Cd sorption capacity increased from 31.9 to 51.4 mg g$^{-1}$ as a result of the increased mineral content in biochar, especially the soluble carbonate (from 2.5% to 2.9%). X-ray diffraction following Cd sorption evidenced that Cd-carbonate and phosphate formed in the biochar (*Zhang et al., 2015*). Moreover, *Trakal et al. (2014)* used Fourier transform-infrared spectroscopy (FT-IR) to follow Cd sorption on biochar with high ash content produced from grape husks and stalks. They suggested that surface precipitation of Cd-carbonate has shifted the peaks of carbonate. A similar mechanism can be found in the sorption of Pb. Formation of Pb-carbonate $Pb_3(CO_3)_2(OH)_2$ and Pb-phosphate $Pb_9(PO_4)_6$ contributed most to the high removal rate of Pb (*Cao et al., 2009*).

## Organic contaminants

It has been proved that biochar produced from biomaterials has favorable removal ability for organic contaminants (*Gwenzi et al., 2017*). In general, pore-filling, hydrophobic effect, electrostatic interaction, and hydrogen bonds are the main mechanisms of organic contaminants sorption by biochar, differing according to the physicochemical properties of the contaminants and biochar.

Pore-filling is an essential mechanism for the sorption of organic compounds on biochar. The sorption capacity is directly in proportion to the micropores' surface area (*Han et al., 2013*). *Chen, Chen & Chiou (2012)* revealed that the biochar's surface area is influenced by the pyrolysis temperature, affecting the uptake rate of naphthalene (NAP) in solutions. The organic components in the biomass were more completely carbonized at higher temperatures, so the biochar had a higher carbonization degree, larger surface area, and more developed micropores, leading to an enhanced sorption rate. Moreover, biochars produced at intermediate temperatures (250−350 °C) displayed relatively slow sorption rates, owing to the difficult pore-filling into certain highly condensed organic phases exited at these temperatures. *Zhu et al. (2014)* reported that the large surface area and pore volume of carbonaceous materials commonly promote the sorption of organic contaminants as a result of the pore-filling effect, which was also verified by research results of *Inyang et al. (2014)* and *Han et al. (2013)*.

*Sun et al. (2013)* explored the influence of deashing treatment on the biochar structure and its sorption ability for phenanthrene (PHE). They reported that after deashing, the hydrophobic domains of biochar increased while the polar functional groups decreased, bringing about more hydrophobic sorption sites for non-polar organic compounds, which promoted PHE sorption. Also, they found that the hydrophobic effect was more significant for biochar prepared at higher temperatures. *Ahmad et al. (2013)* found that there was a more carbonized portion in the biochar produced under high pyrolysis temperature,

resulting in better sorption for relatively hydrophobic trichloroethylene. As pyrolysis temperature increased, the hydrogen- and oxygen-containing functional groups were removed, leading to enhancement of the biochar's hydrophobicity, thus improving the sorption.

Different results also showed electrostatic interaction to be an essential mechanism of polar organic contaminant sorption (*Inyang et al., 2014*). *Xu et al. (2011)* investigated the sorption mechanism of Methyl Violet and found that electrostatic interaction, to be more specific, the attraction between dyes molecules with -COO- and phenolic -OH groups, promoted the sorption of Methyl Violet on biochar. *Xie et al. (2014)* stated that the sorption of sulfonamides (SAs) on different biochars is well correlated with the biochars' graphitization degree and the $\pi - \pi$ electron donor–acceptor (EDA) interaction existed between the graphitic surface ($\pi$ electron donors) and SAs ($\pi$ electron acceptors), accounting for the strong sorption.

*Qiu et al. (2009)* investigated the sorption mechanism of Brilliant Blue (BB) on straw-based biochar. It was suggested that the mechanism involved hydrogen bonds. FT-IR showed that after sorption, the intensity of the peak at 1,795 cm$^{-1}$ reflecting C=O stretching vibration shifted little, and the peak at 3447 cm$^{-1}$ corresponding to -OH stretching vibration had a bit change. There was a good chance that the intermolecular hydrogen bonds (O-H- - -O bonds) existed between the H atom in -OH of BB molecules and the O atom in C=O on biochar surface, vice versa. The negatively charged properties for both biochar and BB also supported this weak interaction.

The co-existence of carbonized and non-carbonized proportions makes the biochar surface heterogeneous; meanwhile, the two types represent different sorption mechanisms. In addition to the sorption of organic compounds onto the carbonized proportion, the partition into the non-carbonized organic matrix is also essential when biochars are produced at lower temperatures (*Zheng et al., 2010*; *Chen, Chen & Chiou, 2012*; *Cao et al., 2009*).

## BIOCHAR APPLICATION IN WATER AND WASTEWATER TREATMENT

Among the increased number of published reports, biochar can be directly used in water and wastewater treatment as a sorbent for contaminants removal, or be used in constructed wetlands (CWs) and in the soil to improve the water quality. Table 1 compiled the references discussed within this section on the removal of various contaminants from water and wastewater by biochar.

### Industrial wastewater

As a dominant source of water contamination, the quantity of industrial wastewater and types of water contaminants are booming due to the rapid development of the industry. Biochar is becoming a new approach to remove various contaminants from industrial wastewater, both for heavy metals and organic compounds.

Removal of $Cd^{2+}$, $Pb^{2+}$, $Cu^{2+}$, $Hg^{2+}$, $Cr^{6+}$, and $Ni^{2+}$ have received much attention due to the adverse effects they could bring if released to the environment. Batch sorption

**Table 1** Removal of various contaminants from water and wastewater by biochar derived from different feedstocks.

| | Biomass feedstock | Production method | Target contaminant | Maximum removal ability | Reference |
|---|---|---|---|---|---|
| Heavy metals | Bamboo, bagasse, hickory wood, peanut hull | Pyrolysis at 600 °C then chitosan modification | $Cd^{2+}$, $Pb^{2+}$, $Cu^{2+}$ | 14.3 mg g$^{-1}$ for $Pb^{2+}$ | *Zhou et al. (2013)* |
| | Malt spent rootlets | Pyrolysis at 850 °C for 1 h | Hg(II) | 103 mg g$^{-1}$ | *Boutsika, Karapanagioti & Manariotis (2014)* |
| | Malt spent rootlets | Pyrolysis at 300–900 °C | Hg(II) | 130 mg g$^{-1}$ for MSR750 | *Manariotis, Fotopoulou & Karapanagioti (2015)* |
| | Waste glue residue | $ZnCl_2$ modification | Cr(VI) | 325.5 mg g$^{-1}$ | *Shi et al. (2020)* |
| | Lotus stalks | Zinc borate as flame retardant, pyrolysis at 300, 350, and 400 °C | Ni(II) | 61.7 mg g$^{-1}$ for 0.5 g ZB/g LS pyrolysis at 300 °C | *Liu et al. (2014)* |
| Dyes | Bamboo cane | Phosphoric acid modification then pyrolysis at 400, 500, and 600 °C | Lanasyn Orange and Lanasyn Gray | $2.6 \times 10^3$ mg g$^{-1}$ for both dyes | *Pradhananga et al. (2017)* |
| | Pecan nutshell | Pyrolysis at 800 °C for 1 h | Reactive Red 141 | 130 mg g$^{-1}$ | *Zazycki et al. (2018)* |
| Phenols and PAHs | Sewage sludge | Pyrolysis at 500 °C for 1 h/microwave-assisted pyrolysis at 980 W for 12 min | Hydroquinone | 1,218.3 mg g$^{-1}$/1,202.1 mg g$^{-1}$ | *dos Reis et al. (2016)* |
| | Malt spent rootlets | Pyrolysis at 800 °C for 1 h | Phenanthrene | 23.5 mg g$^{-1}$ | *Valili et al. (2013)* |
| | Orange peel | Pyrolysis at 150–700 °C for 6 h | Naphthalene and 1-naphthol | 80.8 mg g$^{-1}$ for naphthalene and 186.5 mg g$^{-1}$ for 1-naphthol | *Chen & Chen (2009)* |
| Pesticides | Maize straw and pig manure | Pyrolysis at 300, 500, and 700 °C for 4 h | Thiacloprid | About 8.1 mg g$^{-1}$ | *Zhang et al. (2018)* |
| | Almond shell | Pyrolysis at 650 °C for 1 h with steam activation at 800 °C | Dibromochloropropane | 102 mg g$^{-1}$ | *Klasson et al. (2013)* |
| | Broiler litter | Pyrolysis at 350 and 700 °C with and without steam activation at 800 °C | Deisopropylatrazine | About 83.3 mg g$^{-1}$ for BL700 with steam activation | *Uchimiya et al. (2010)* |
| | Maple, elm and oak woodchips and barks | Pyrolysis at 450 °C for 1 h | Atrazine and simazine | 451–1,158 mg g$^{-1}$ for atrazine and 243–1,066 mg g$^{-1}$ for simazine | *Zheng et al. (2010)* |
| Antibiotics | Sawdust | $ZnCl_2$ and $FeCl_3$ $6H_2O$ solution doped at 100 °C then calcined at 600 °C for 2 h | Tetracycline | Above 89% after three cycles | *Zhou et al. (2017)* |
| | Potato stems and leaves | Magnetization then humic acid-coated | Fluoroquinolones | 8.4 mg g$^{-1}$ for ENR, 10.0 mg g$^{-1}$ for NOR, and 11.5 mg g$^{-1}$ for CIP | *Zhao et al. (2019)* |

Wang et al. (2020), *PeerJ*, DOI 10.7717/peerj.9164

**Table 1** (*continued*)

| | Biomass feedstock | Production method | Target contaminant | Maximum removal ability | Reference |
|---|---|---|---|---|---|
| Indicator organisms and pathogens | Rice husk | Pyrolysis | Fecal indicator bacteria | 3.9 log units of bacteria removed | *Kaetzl et al. (2019)* |
| | Hardwood | Pyrolysis | *Saccharomyces cerevisiae* | >1 $\log_{10}$ CFU of bacteria removed | *Perez-Mercado et al. (2019)* |
| | Wood chips | Pyrolysis with steam activation | *Escherichia coli* | $3.62 \pm 0.27$ log units of bacteria removed | *Mohanty et al. (2014)* |
| | Bamboo | Pyrolysis at 370 °C | $NH_4^+$ | 6.4 mM $g^{-1}$ | *Fan et al. (2019)* |
| | Bamboo | Pyrolysis at 460 °C/immersed in clay suspension then pyrolysis at 460 °C | $NO_3^-$ | 5 mg $g^{-1}$/9 mg $g^{-1}$ | *Viglašová et al. (2018)* |
| | Walnut shell and sewage sludge | Pyrolysis at 600 °C for 3 h with different ratios of the two feedstocks | $PO_4^{3-}$ | 303.5 mg $g^{-1}$ for pure sewage sludge biochar | *Yin, Liu & Ren (2019)* |
| Inorganic ions | Wood and rice husks | Magnetic modification by co-precipitation of Fe(II)/Fe(III) ions | $PO_4^{3-}$ | 25-28 mg $g^{-1}$ | *Ajmal et al. (2020)* |
| | Spruce wood | Impregnated with $AlCl_3$/$FeCl_3$ solution then pyrolysis at 650 °C for 1 h | $F^-$ | 13.6 mg $g^{-1}$ | *Tchomgui-Kamga et al. (2010)* |

experiments by *Zhou et al. (2013)* showed that the biochar modified by chitosan had favorable removal efficiency for three heavy metals ($Cd^{2+}$, $Pb^{2+}$, and $Cu^{2+}$) from solutions. Further research of $Pb^{2+}$ sorption implied that the biochar had a comparatively high Langmuir sorption capacity of 14.3 mg $g^{-1}$, despite the slow sorption kinetics. *Boutsika, Karapanagioti & Manariotis (2014)* employed biochar produced from malt spent rootlets (MSR) to remove Hg(II) from pure aqueous solutions. The removal efficiency was up to 100% after a 24 h contact time at biochar concentrations of 1 g $L^{-1}$, with the maximum Hg(II) sorption capacity of 103 mg $g^{-1}$. Their later study showed that the Hg sorption capacity by MSR biochars increased by a maximum factor of 6 for high-temperature (750−900 °C) biochars compared to the raw material (*Manariotis, Fotopoulou & Karapanagioti, 2015*). Hg(II) sorption onto both materials carried on mainly through neutral species (*Boutsika, Karapanagioti & Manariotis, 2017*). *Shi et al. (2020)* developed a $ZnCl_2$-modified glue residue biochar for Cr(VI) sorption. The maximum sorption capacity reached 325.5 mg $g^{-1}$, higher than the previously reported sorbents. *Liu et al. (2014)* used zinc borate (ZB) as a flame retardant to prepare lotus stalks (LS) biochar for Ni(II) removal. Sorption of Ni(II) on LS biochar was enhanced by 3–10 times compared with that of biochar without adding ZB.

With the textile industry expanding rapidly, dye wastewater now accounts for a large proportion of industrial wastewater. Among the methods of dye wastewater treatment, biochar sorption is especially favored. For example, *Pradhananga et al. (2017)* reported that two dyes used in wool carpet dyeing, Lanasyn Orange and Lanasyn Gray, could be highly sorbed on nanoporous biochar derived from bamboo cane. The sorption capacity of both dyes was $2.6 \times 10^3$ mg $g^{-1}$, assuming pore-filling to be the primary sorption mechanism, and the high sorption capacity was attributed to the high specific surface area (2,130 $m^2$ $g^{-1}$) and pore volume (2.7 $cm^3$ $g^{-1}$) of biochar. Researchers produced pecan nutshell biochar to remove Reactive Red 141 from water. The biochar was claimed to be low-cost and environmentally friendly, which could be a substitute for other conventional sorbents (*Zazycki et al., 2018*).

Emerging organic contaminants in industrial wastewater, such as phenols and PAHs, have gained great concern. *dos Reis et al. (2016)* produced biochar from sewage sludge by pyrolysis at 500 °C, followed by HCl treatment. The biochar displayed a very high sorption capacity for hydroquinone, which was up to 1218.3 mg $g^{-1}$. $\pi - \pi$ EDA interactions play significant roles in the sorption. *Valili et al. (2013)* reported that the MSR biochar pyrolyzed at a higher temperature of 800 °C gained a much higher PHE sorption capacity, two orders of magnitude higher compared to the raw material. *Chen & Chen (2009)* made orange peel biochar with a pyrolysis temperature ranging from 150 to 700 °C (OP150-OP700) for sorption of 1-naphthol and NAP. For biochars pyrolyzed at lower temperatures, their polar surface due to the presence of water molecules has additional polar interactions (e.g., hydrogen bonds) with 1-naphthol, resulting in higher sorption capacity than NAP. Meanwhile, the partition is favored as the sorbate concentration increases, but adsorption rapidly reaches saturation (*Chen, Zhou & Zhu, 2008*). OP200 had the maximal sorption capacity for 1-naphthol with high concentration due to polar interactions and high

partition. However, OP700 exhibited an optimum sorption capacity for NAP because of its highest surface area and low surface polarity, which facilitated the NAP sorption.

## Pesticides

Utilization of pesticides benefits the agricultural production and the economy, but excessive use of pesticides causes toxicity on non-target organisms and destruction to ecological balance and human health (*Zhong et al., 2018*). Biochar is applied as a distinctive remediation method in pesticide contamination treatment (*Dai et al., 2019*).

*Zhang et al. (2018)* produced maize straw biochar at 300, 500, and 700 °C to study thiacloprid sorption. They found that the sorption occurred probably via pore-filling, hydrophobic interaction, and $\pi - \pi$ interaction. *Jin et al. (2016)* prepared biochar by pyrolysis of swine manure at 600 °C, which was used for imidacloprid sorption. The results showed that pore-filling is likely one of the dominant sorption mechanisms for this kind of polar chemical. *Klasson et al. (2013)* prepared almond shell biochar by pyrolysis with steam treatment. The biochar had a larger specific surface area of 344 m$^2$ g$^{-1}$ and a sorption capacity of 102 mg g$^{-1}$ for dibromochloropropane, a nematode insecticide, and the field experiment was carried out successfully.

*Zheng et al. (2010)* investigated the sorption of two triazine pesticides, atrazine and simazine on biochar. Based on different sorption conditions, the sorption ability of atrazine was 451–1158 mg g$^{-1}$, and 243–1066 mg g$^{-1}$ for simazine. When the two sorbates existed synchronously, there was competitive sorption on biochar. The sorption capacity of atrazine was 435–286 mg g$^{-1}$, and 514–212 mg g$^{-1}$ for simazine. The study also reported that the sorption process of both single and multiple triazine pesticides on biochar could be well explained by surface sorption mechanism. *Uchimiya et al. (2010)* produced broiler litter biochar by pyrolysis at 350 and 700 °C to remove deisopropylatrazine, a stable metabolite of atrazine from water. They found that the biochar prepared at 700 °C had higher surface area, more micropores in non-carbonized fraction, and greater aromaticity. Thus, the target contaminant can be effectively removed, while the removal efficiency of biochar prepared below 500 °C was relatively low.

## Antibiotics

Some antibiotics in pharmaceutical wastewater are difficult to decompose in the natural environment and regarded as emerging environmental contaminants (*Carvalho & Santos, 2016*). Reducing the toxicity of antibiotics by biochar becomes a hot spot.

Tetracyclines (TCs) and SAs are two of the most commonly used antibiotics and are also used in intensive agriculture as feed additives, bringing potential hazards to the environment and human health when extensively used (*Yu et al., 2016*; *Shao et al., 2005*). The removal of TCs by ZnCl$_2$/FeCl$_3$ solution doped sawdust biochar was studied systematically. Results showed that this kind of biochar had the potential ability for TCs removal in water, with the removal rate above 89% after three cycles (*Zhou et al., 2017*). *Peiris et al. (2017)* made a further study on the sorption mechanisms of SAs on biochar. Generally, high-temperature produced biochar showed high sorption quantity under the condition of weak acidity, attributed to strong $\pi - \pi$ EDA interactions between the abundant arene rings on the

biochar surface and SAs molecules. Micropore-filling is also a common mechanism because of the smaller size of SAs. *Zhao et al. (2019)* prepared humic acid-coated magnetic biochar derived from potato stems and leaves to sorb three typical fluoroquinolones (FQs)—enrofloxacin (ENR), norfloxacin (NOR), and ciprofloxacin (CIP). The maximum adsorption capacities were 8.4 mg g$^{-1}$ for ENR, 10.0 mg g$^{-1}$ for NOR, and 11.5 mg g$^{-1}$ for CIP. High FQs removal efficiency could be owing to hydrophobic, electrostatic and $\pi - \pi$ EDA interactions and formation of hydrogen bonds.

## Indicator organisms and pathogens

Biochar application in the removal of indicator organisms and pathogens mainly aims at the treatment of urban stormwater runoff, which contains a wide range of contaminants and eventually runs into surface water (e.g., streams, lakes). Irrigation with these contaminated waters can lead to microbial contamination of vegetables. Biochar filters for microbe removal from water have received considerable attention.

*Kaetzl et al. (2019)* studied the filtration of rice husk biochar and non-pyrolyzed rice husk as low-cost filter materials for wastewater and evaluated their potential and limitation. In general, the performance of the biochar filter was superior or equal to the rice husk and standard sand filters. The treated wastewater was then used in a pot test for lettuce irrigation. Results showed that the contamination with fecal indicator bacteria was >2.5 log units lower than the control group irrigated with untreated wastewater. Mechanisms responsible for the removal include the filtration of larger pathogens and the sorption of negatively charged bacterial and cells (*Gwenzi et al., 2017*). Similarly, *Perez-Mercado et al. (2019)* showed that by using biochar as a filter medium, >1 log$_{10}$ CFU *Saccharomyces cerevisiae* was successfully removed from diluted wastewater under the condition of on-farm irrigation. The particle size of biochar is the main influencing factor accounting for the microbial removal efficiency. The minimum particle size (d$_{10}$ = 1.4 mm) could consistently remove at least 1 log$_{10}$ CFU of most target microbes. More micropores and smaller pore size of biochar could increase straining effect and contact time between bacteria and sorption sites. *Mohanty et al. (2014)* improved sand biofilters with 5 wt% biochar amended to increase the bacteria removal capacity. The biochar-amended sand filter retained up to three orders of magnitude more *Escherichia coli* and prevented their mobility during continuous, intermittent flows. The improved removal capacity of pathogens was attributed to higher retention on the biochar filter, which increased the attachment of *E. coli*.

## Inorganic ions

In virtue of the convenience and little generation of secondary contamination (*Yang et al., 2018*; *Yin et al., 2017*), biochar are popular in inorganic ions removal, which targets at the removal of nutrient elements N and P that exist in the form of inorganic ions in wastewater, and F$^-$ in drinking water.

*Fan et al. (2019)* conducted a study on NH$_4^+$ sorption by hydrous bamboo biochar. Results found that the biochar had a sufficient sorption capacity for NH$_4^+$, with a maximum of 6.4 mM g$^{-1}$. The sorption was enhanced at higher ionic strength conditions, indicating that physical reactions possibly made contributions to the sorption process

such as electrostatic interactions. Potential mechanisms for $NH_4^+$ sorption was further studied by *Hu et al. (2020)*. They reported that pH influenced the $NH_4^+$ sorption capacity by changing the surface charge of biochar. Negatively charged biochar in higher pH (pH > $pH_{PZC}$) solutions easily sorbed $NH_4^+$ due to electrostatic attraction. FT-IR patterns showed that the -OH and -C=O groups weakened after the sorption, indicating that the $NH_4^+$ acted with these functional groups through surface complexation. In addition, ion exchange between $NH_4^+$ and the negatively charged functional groups such as -OH and -COOH also led to the $NH_4^+$ sorption. For $NO_3^-$, the sorption mechanisms are governed by multiple interactions, primarily electrostatic attraction, and ionic bonds with exchangeable cations from the biochar, based on the sorption study of $NO_3^-$ by bamboo biochar (*Viglašová et al., 2018*).

Walnut shell and sewage sludge were co-pyrolyzed to prepare biochar for $PO_4^{3-}$ sorption from eutrophic water (*Yin, Liu & Ren, 2019*). The biochars exhibited ideal sorption ability, among which the pure sewage sludge biochar had the maximum sorption capacity of 303.5 mg $g^{-1}$ in a wide pH range and was the best option for $PO_4^{3-}$ sorption among the biochars. *Ajmal et al. (2020)* compared the removal efficiency of $PO_4^{3-}$ from wastewater by biochars before and after magnetic modification. Results showed that the sorption ability of magnetic biochar was twice (25–28 mg $g^{-1}$) than that of the unmodified biochar (12–15 mg $g^{-1}$). The $PO_4^{3-}$ sorption on magnetic biochar is dominated by simultaneous mechanisms including electrostatic attraction, surface precipitation, and complexation, while for the original biochar, the sorption mainly depends on electrostatic attraction.

$F^-$ is characterized by high electronegativity and small ionic size, resulting in a strong affinity towards metal ions such as Al(III), La(III), and Fe(III) (*Wu et al., 2007*). Thus, a strong $F^-$ sorption could be achieved by composites made of such metal ions dispersing in a porous matrix such as biochar. Such a study was made by *Tchomgui-Kamga et al. (2010)*, which found that the Al-modified spruce wood biochar had a maximum removal capacity of 13.6 mg $g^{-1}$ for $F^-$. The dispersion of Al into the porous structure of biochar significantly increased the sorption. The Langmuir isotherm model served as the most suitable model for $F^-$ sorption (*Ahmed et al., 2016*).

## Indirect water and wastewater treatment

In recent years, CWs have been widely used in wastewater treatment, including removal of N, P (*Li et al., 2019*), and some organic contaminants. Nevertheless, due to restricted oxygen supply and transport capacity, limited sorption capacity of the substrate, and inhibition of microbes and plants metabolism at low temperatures, the removal efficiency for N and P is severely hindered (*Ying et al., 2010*). Researchers have attempted to explore particular substrates to intensify the functions of CWs with high contaminants concentration, among which biochar has been favorably considered (*Gupta, Prakash & Srivastava, 2015*).

*Zhou et al. (2018)* used biochar as a substrate in vertical flow constructed wetlands (VFCWs) to enhance the removal efficiency with a series of low C/N ratio influent strengths. They assessed the removal of N and organic contaminants in both VFCWs with/without biochar added. Results showed that the average removal rates of $NH_4$-N (39%), TN (39%), and organic contaminants (85%) were better than those of conventional VFCWs,

especially for the high-strength wastewater. A seven-month study by *Bolton et al. (2019)* clearly showed that enriched biochar was a suitable substrate for $PO_4$-P removal. The waste biochar has the potential for regeneration, which can be applied as soil fertilizer to improve soil quality, while this application still needs more investigations. *Deng et al. (2019)* set up four subsurface flow constructed wetlands (SFCWs) with biochar amended in standard gravel at different volume ratios (0–30%). Results indicated that the removal rates of $NH_4$-N and TN by SFCWs with biochar were higher than those by pure gravel-filled SFCWs. The additive of biochar promotes N removal by changing the structure of microbial communities and increasing the abundance of dominant species. Besides, biochar improves the metabolism of high molecular compounds and convert them into low molecular compounds. These results provide new insights into strengthening N removal through microbial metabolism with the effect of biochar.

Surface runoff and soil erosion in the river basin, especially in some degraded fields with high precipitation, could cause certain contamination to the water environment. Several studies have proved that biochar has the potential to reduce surface runoff and soil erosion (*Razzaghi, Obour & Arthur, 2020*; *Gb et al., 2020*; *Bayabil et al., 2015*; *Tanure et al., 2019*; *Gholami, Karimi & Kavian, 2019*). Biochar particles can bond with soil mineral surface through phenolic and carboxylic functional groups, thus improve the stability of soil aggregation and structure (*Soinne et al., 2014*). Besides, the exchangeable divalent cations with high charge density (e.g., $Ca^{2+}$, $Mg^{2+}$) on biochar surface can replace the monovalent cations (e.g., $Na^+$, $K^+$) on exchange sites of clay particles, which enhances clay flocculation and thereby improves macropores size and network in the soil (*Rao & Mathew, 1995*), eventually increases the infiltration capacity. Therefore, it is concluded that the biochar amendment can improve soil physical properties, which in turn reduces runoff, erosion, and waterlogging (*Bayabil et al., 2015*). Moreover, biochar with large water-storing property spreading on soil surface could absorb the force of raindrops, thus increases the runoff time (*Gholami, Karimi & Kavian, 2019*).

## Current application of biochar in wastewater treatment facilities

Although biochar exhibits some similar properties as the activated carbon, it is a more heterogeneous material with many uncertainties when applied in engineered facilities (*Gwenzi et al., 2017*). Situations involving ion strength, pH, or presence of organic matters make the sorption more complex. Compared with current wastewater treatment facilities with mature technologies, which usually use activated sludge, activated carbon, and a series of water treatment agents such as flocculants and disinfectants, there are limited attempts to develop biochar-based wastewater treatment facilities. Despite the published efforts on the removal of various contaminants by biochar, the studies are based on laboratory batch experiments. Operation parameters and conditions for real facilities remain lacking.

To date, biochar-based filters have been an attempt to advance the engineered application of biochar. Sand filters and biofilters amended with biochar (*Kaetzl et al., 2019*; *Perez-Mercado et al., 2019*), and filters made of biochar-clay composite (*Chaukura et al., 2020*), all have shown the improvements in wastewater treatment performance. Notably, a pilot-scale biochar-based wastewater treatment system called

N-E-W Tech[TM] was built and patented by Greg Möller from the University of Idaho in 2015 (https://www.lib.uidaho.edu/digital/uinews/item/n-e-w-tech-project-proposes-better-water-treatment-system.html). This system promises highly efficient removal of phosphorus and mineral contaminants from wastewater; meanwhile, it makes use of the minerals stripped from water to produce fertilizer, which is also cost-effective. The system was then licensed and promoted in real wastewater treatment systems in the USA, England, and South Korea. This case demonstrates the scalability of biochar engineered application and provides guidance as well.

## BIOCHAR MODIFICATION

Although biochar has been extensively applied in the removal of diversiform contaminants in water solutions, its applicability is limited because of the lower removal efficiency for some selected contaminants or in some specific water conditions. The unmodified biochars have much lower removal ability than the modified ones, especially in high-strength wastewater (*Rangabhashiyam & Balasubramanian, 2019*). Researchers have found relationships between the surface area and functionality of biochar with the sorption capacity (*Tan et al., 2015*; *Goswami et al., 2016*). More micropores and mesopores correspond to larger surface area and more sorption sites where contaminants can be sorbed (*Sizmur et al., 2017*). Accordingly, the modification of biochar generally concerns (i) increasing the surface area and porosity; (ii) enhancing the surface properties; (iii) embedding other materials into the biochar matrix to obtain beneficial composites (*Sizmur et al., 2017*). According to different modification emphases, modification methods of biochar are summarized in Fig. 2.

### Increasing surface area and porosity

In general, biochar with larger surface area has more sorption sites, facilitating the sorption capacity. Plenty of modification methods of biochar have been proposed to achieve this favorable property.

Physical modification usually uses gases such as $CO_2$ (*Guo et al., 2009*) and steam (*Shim et al., 2015*) to treat biochar at the temperature over 700 °C. With steam treatment, the incomplete combustion components are removed, and the porosity is improved, both of which increase the sorption sites. *Lima & Marshall (2005)* pyrolyzed poultry manure at 700 °C to produce biochar, followed by a series of steam with different water flow rates and durations at 800 °C. Results showed that longer action times and higher flow rates increased the sorption of Zn, Cu, and Cd on the biochar surface. *Zhang et al. (2004)* investigated the effect of $CO_2$ treatment duration on biochars derived from corn stover, corn hulls, and oak wood waste. All biochars exhibited higher sorption capacity with longer treating duration owing to the larger surface area and micropore volume. *Kangyi Lou (2016)* claimed that the steam treatment had no significant effect on surface functional groups on biochar. Therefore, the steam treatment appears to be more efficient if it is used before a second

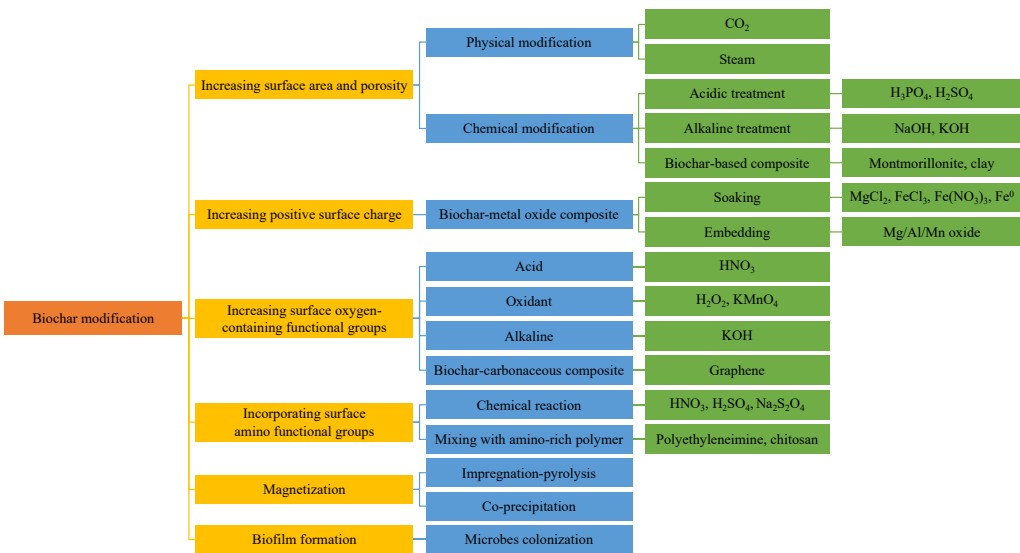

**Figure 2 Modification methods of biochar according to different emphases.**

modification step, which can increase the number of surface functional groups (*Sizmur et al., 2017*).

Acidic or alkaline treatment also increases the surface area. *Zhao et al. (2017)* treated pine tree sawdust with diluted $H_3PO_4$ before pyrolysis. Both the total surface area and pore volume increased after the treatment, and the sorption capacity for Pb increased by more than 20% because of surface sorption and phosphate precipitation. *Goswami et al. (2016)* proved that pyrolyzing the biochar-KOH mixture at 350−550 °C reopened some of the blocked pores, and expanded the pore size of smaller pores, increasing the surface area and Cd sorption from the water via surface complexation. *Hamid, Chowdhury & Zain (2014)* reported that the increase of surface area resulting from KOH modification also increased the sorption of oxyanions. For that, *Jin et al. (2014)* proposed that the maximum As(V) sorption on biochar modified by KOH increased from 24 mg g$^{-1}$ to 31 mg g$^{-1}$, as a result of increased surface area. Similarly, researchers found a larger surface area and iodine sorption capacity of both the feedstock and biochar when the modification was conducted by mixing with solid NaOH (*Pietrzak et al., 2014*).

Except for the physical, acidic, and alkaline modification mentioned above, some of the biochar-based composites also possess a larger surface area by impregnating biochar with specific materials. In this case, the biochar primarily plays a role as a scaffold with high surface area on which other materials are deposited (*Sizmur et al., 2017*). *Chen et al. (2017)* pointed out that the additive of montmorillonite during the pyrolysis of bamboo powder led to an increase in surface area and porosity, partially as a result of the existence of layered montmorillonite, which contributed to better sorption capacities for $NH_4^+$ and $PO_4^{3-}$. *Yao et al. (2014)* observed the layered surface of clay modified biochar through scanning electron microscope (SEM), similar to a typical clay structure morphology.

## Increasing positive surface charge

Generally, the surface charge of biochar is negative and has a higher pH value, making biochar an excellent sorbent of metal cations, while a poor sorbent for oxyanions such as $NO_3^-$, $PO_4^{3-}$, and $AsO_4^{3-}$ (*Sizmur et al., 2017*). Thus, modifications usually use the porous surface of biochar as a scaffold for embedding positively charged metal oxides. The obtained composites can remove oxyanions with negative charge from water (*Sizmur et al., 2017*).

Most methods to prepare biochar-metal oxide composites aim to assure the homogeneous distribution of metals on biochar surface. Biochar here plays a role as porous carbon support where the metal oxides precipitate to gain more positive surface charge and surface area simultaneously. In general, biochar or raw materials soaked into metal chloride or nitrate solutions ($MgCl_2$, $FeCl_3$, and $Fe(NO_3)_3$) are most frequently used to realize the attachment of metals. After heating under atmospheric conditions at $50-300\ °C$, the chlorides or nitrates were driven off as $Cl_2$ and $NO_2$ gases, and the metal ions were converted into metal oxides (*Sizmur et al., 2017*). *Zhang et al. (2012)* used several common biomass wastes to create biochar-MgO composites by mixing the feedstocks with $MgCl_2 \cdot 6H_2O$ solution and then pyrolyzing. SEM images showed that MgO particles were uniformly spread on the biochar surface. The maximum sorption capacity for nitrogen and phosphorus from sewage reached 95 and 835 mg g$^{-1}$, respectively, due to positively charged MgO that precipitated onto the biochar. They also produced biochar/MgAl-layered double hydroxides by mingling cotton stalk with a mixed solution of $AlCl_3 \cdot 6H_2O$ and $MgCl_2 \cdot 6H_2O$ (*Zhang et al., 2013*). The maximum sorption capacity for phosphorus increased by 5–50 times.

Embedding Mg, Al, or Mn oxides onto the biochar surface also produces biochar-based composites, which can improve the sorption of both metal cations and oxyanions in water solutions. *Jellali et al. (2016)* explored the effects of Mg modification on sorption ability for metal cations. In this study, Pb sorption by a cypress sawdust-derived and $MgCl_2$-treated biochar was investigated. Results showed that the modified biochar obtained an enhanced sorption capacity, about 7.4 times more compared to the raw material.

In general, the sorption of oxyanions by biochar-metal oxide composites arises from electrostatic attraction or chemical sorption with positively charged metal oxides in the biochar matrix (*Zhou et al., 2014*; *Ren et al., 2015*), while the sorption of metal cations is caused by co-precipitation occurring in the metal oxides lattice, or chemical sorption on oxygen-containing functional groups on the biochar's unmodified part (*Tan et al., 2015*). *Rajapaksha et al. (2016)* suggested that even though most modifications by metal oxides decreased the surface area because of pore-clogging with metal oxide precipitates, the modifications eventually increased the sorption capacity owing to the formation of pH-dependent bonds with positively charged functional groups on the biochar surface.

## Increasing surface oxygen-containing functional groups

The biochar surface contains several functional groups such as carboxyl, hydroxyl, and phenolic groups, which are capable of chemically binding with contaminants and remove them from aqueous solutions.

The acidic treatment provides additional oxygen-containing functional groups on the biochar surface and increases the potential of chemically binding with positively charged contaminants via specific sorption. The biochar forms carboxylic groups on its surface when exposed to acidic solutions (*Qian et al., 2013*; *Hadjittofi, Prodromou & Pashalidis, 2014*). *Hadjittofi, Prodromou & Pashalidis (2014)* used $HNO_3$ to modify biochar produced from cactus fibers to obtain more surface carboxylic groups as sorption sites for metal cations ($Cu^{2+}$ and $Pb^{2+}$). The sorption capacity at pH 6.5 was an order of magnitude larger than that at pH 3, indicating the pH-dependent and chemical sorption on oxygen-containing functional groups. *Qian et al. (2013)* suggested that after the treatment in a mixture of $H_2SO_4$ and $HNO_3$, the O/C ratio of rice straw biochar was higher in the final product, implying that oxygen-containing functional groups were enriched in the structure of biochar.

Since biochar modification by strong acids is costly in a large-scale application and causes environmental concerns when disposing of the modification agents, researchers have made efforts to come up with cheaper and cleaner oxidants as alternatives to modify biochar. *Song et al. (2014)* pyrolyzed corn straw at 600 °C and then mixed it with $KMnO_4$ solution. A $MnO_x$-biochar was prepared after another pyrolysis. Compared with the original biochar, the O/C ratio increased from about 0 to 0.5. XPS analyses showed that the increased oxygen existed mainly in the Mn-OH and Mn-O structure, which primarily accounted for the enhanced sorption ability for $Cu^{2+}$ (from 19.6 to 160.3 mg g$^{-1}$). *Huff & Lee (2016)* showed an increased number of oxygen-containing functional groups on the biochar surface after treatment using $H_2O_2$. The cation exchangeability of the biochar was almost doubled than that of the untreated one, as a result of cation exchange on the more abundant oxygen-containing functional groups on the modified biochar surface.

Alkaline solutions play a similar role to acids and oxides in increasing the number of oxygen-containing functional groups on the biochar surface. *Jin et al. (2014)* reported that KOH modification of biochar made of municipal solid wastes enhanced the As(V) sorption performance, not only because of the increased surface area but also the growing number of surface oxygen-containing functional groups, which provided proton-donating exchange sites where metal cations can be chemically sorbed (*Petrović et al., 2016*).

Among various biochar-based composites, the biochar-graphene oxide composite material, which is obtained by impregnating the raw material in a graphene oxide suspension and then pyrolyzing, also displays more oxygen-containing functional groups after incorporating the graphene structure (*Tang et al., 2015*; *Shang et al., 2016*). The removal rate of $Hg^{2+}$ raised with the increase of the proportion of graphene oxide in the composite. When the maximum percentage of graphene oxide is 1%, the removal rate of the composite was 8.7% more than that of the unmodified biochar. FT-IR showed that the abundant oxygen-containing functional groups dominate the sorption behavior of $Hg^{2+}$ on the biochar-graphene oxide composite.

## Incorporating surface amino functional groups

Incorporating amino functional groups onto the biochar surface improves the sorption ability through inducing strong complexation between contaminants and the amino

sites. The modification is obtained either by chemical reactions or blending biochar with amino-rich polymers such as polyethyleneimine (PEI) and chitosan (*Zhou et al., 2013*; *Zhou et al., 2014*; *Yang & Jiang, 2014*).

*Yang & Jiang (2014)* used $HNO_3$, $H_2SO_4$, and $Na_2S_2O_4$ to modify biochar via nitration and reduction reactions as a selective and efficient sorbent for $Cu^{2+}$. Although there was little significant difference in the physical structure before and after the modification, attenuated total reflectance FT-IR and XPS results showed that the amino groups chemically bound with the functional groups on the biochar surface. The amino modification made the sorption capacity for $Cu^{2+}$ increased by five times. *Ma et al. (2014)* used PEI to prepare amino-rich biochar to remove Cr(VI) from aqueous solutions, which obtained a much higher maximum sorption capacity ($435.7$ mg g$^{-1}$) than that of the unmodified biochar ($23.1$ mg g$^{-1}$).

*Zhou et al. (2013)* synthesized chitosan-modified biochars derived from peanut hull, hickory wood, sugarcane bagasse, and bamboo, aiming to provide a commercial sorbent for heavy metals remediation in the water environment. Characterization of the biochars showed that the chitosan coating on the biochar surface improved the surface properties. Batch sorption experiments stated that the removal abilities for $Cd^{2+}$, $Cu^{2+}$, and $Pb^{2+}$ in aqueous solutions by almost all chitosan-modified biochars were enhanced, compared with the unmodified biochars. Further studies of Pb sorption on chitosan-modified bamboo biochar found that, even though the sorption kinetics were slow, the modified biochar had a relatively high Langmuir Pb sorption capacity of $14.3$ mg g$^{-1}$, significantly reducing the toxicity of Pb. Characterization of the Pb-loaded biochar after sorption exhibited that the sorption of Pb is primarily caused by the interaction with amino functional groups on the biochar surface.

## Magnetization

The magnetization of biochar is a new modification frontier. It develops in situations where the separation of biochar from aqueous solutions face great difficulties. The application of a magnet for magnetic biochar enables such difficulty to be solved.

Impregnation-pyrolysis and co-precipitation are the most commonly used preparation method for magnetic biochar, accounting for about 69.6% of all preparation methods (*Yi et al., 2020*). Impregnation-pyrolysis is to impregnate the suspension of biochar with a solution of transition metal salts, followed by pyrolysis of the residue. In this way, *Mohan et al. (2014)* produced magnetic biochar using $Fe^{3+}/Fe^{2+}$ solution. It was found that the iron content increased from 1.4% to 80.6%, indicating that the biochar was effectively magnetized. In the application of $Pb^{2+}$ and $Cd^{2+}$ removal from solutions, the biochar showed significantly higher sorption capacity. Except for conventional pyrolysis, microwave heating is extensively applied in the synthesis of magnetic biochar. *dos Reis et al. (2016)* produced biochars in different methods—pyrolysis at 500 °C and microwave heating under an inert atmosphere—both were followed by HCl treatment. The biochars had approximately equal and very high sorption capacity for hydroquinone, showing that microwave heating could be an alternative to conventional pyrolysis. The co-precipitation synthetic pathway includes the dispersion of biochar in a solution of transition metal salts,

adjusting the pH to 9–11 with NaOH or ammonia solution with constant stirring. Magnetic biochar is obtained by drying the residue (*Yi et al., 2020*). *Yu et al. (2013)* obtained magnetic biochar by mixing $Fe^{2+}/Fe^{3+}$ solution into an ammonia solution with biochar particles dispersed, followed by ultrasound irradiation at 60 °C. The magnetic biochar exhibited an increased number of carboxyl functional groups on the surface, resulting in a more negatively charged property, which improved the sorption rate and capacity for heavy metal ions.

In addition to surface functional groups that take effects in the sorption process, the magnetic components which exist in the main forms of $Fe_2O_3$, $Fe_3O_4$, FeO, and $Fe^0$, also play an important role in improving the sorption ability (*Yi et al., 2020*). For example, $Fe^0$ makes vital contributions to Pb(II) removal by directly reduction (*Chen et al., 2018*), while $Fe_3O_4$ plays a crucial role in remediation of Cr(VI), attributed to the Fe(II) and Fe(III) in octahedral coordination in $Fe_3O_4$, which act as active chemical sorption or reduction sites (*Zhong et al., 2018*). Synthetic conditions such as pyrolysis temperature influence the morphology of magnetic components, for instance, the $Fe_3O_4$ in magnetic biochar transformed into FeO when the pyrolysis temperature increased (*Chen et al., 2019*). Moreover, innovative synthetic methods that introduce other metals such as Cu, Zn, and Mn lead to the formation of magnetic substances containing these metals, playing a particular role in enhancing the removal effect (*Zhang et al., 2019*; *Heo et al., 2019*).

## Biofilm formation

Taking advantage of the high surface area, porosity, and inert property, biochar can be used as a scaffold for colonization and growth of biofilms. The microbes adhere to the biochar surface and develop an extracellular biofilm by secreting multiple polymers as an adhesive, and therefore have stronger viability owing to the protection from the biofilm, excelling the traditionally separate microbial treatment (*Hall-Stoodley, Costerton & Stoodley, 2004*). In such biotic systems, biochar plays its role in the sorption of contaminants by the porous structure and surface functional groups, while the microbes promote the degradation of resistant compounds owing to their metabolism (*Singh, Paul & Jain, 2006*). The synergistic removal effect makes such biotic biochar be increasingly used in water and wastewater treatment.

The primary purpose of biochar with biofilm is to promote the biodegradation of organic contaminants (*Sizmur et al., 2017*). *Dalahmeh et al. (2018)* studied the potential of biochar filters with biofilm as a substitution or progress of conventional sand filters for contaminants removal from pharmaceutical wastewater. For carbamazepine, the biotic biochar possessed more effective and stable removal efficiency than sand filters, more than 98% over the 22 weeks of operation. The combination of sorption and simultaneous biodegradation are conducive to the removal. *Frankel et al. (2016)* proved the synergistic behavior by biofilm and biochar in naphthenic acid (NA) removal from water solutions. The biotic biochar had a higher NA removal rate (72%) than either the sterile biochar (22–28%) or the microbes alone (31–43%). Interestingly, in the presence of metals (Al and As), although there was a reduction in the microbial proliferation, the removal of NA by the biochar-biofilm coalition increased to 87%. An enhancement in metal sorption

was also observed, indicating a synergistic removal in the co-existence of organic and inorganic contaminants. The results suggest a biochar-biofilm combined approach to treating co-contaminated industrial wastewater, though the removal mechanisms need to be further studied.

All in all, the selection of modification methods should base on the property and removal mechanism of the target contaminant. Generally, gas, steam, acid, and alkaline modifications increase the porosity of biochar, suitable for the contaminant whose sorption is dominant by pore-filling. Both acids and oxidants agents enrich the surface oxygen-containing groups of biochar with high cation exchangeability, which facilitates the sorption governed by ion exchange, such as heavy metal ions and $NH_4^+$. The alkaline modification provides high aromaticity of the biochar, promoting the $\pi - \pi$ EDA interaction and sorption for some organic contaminants such as dyes and antibiotics. Notably, it also leads to a lower O/C ratio, which strengthens the hydrophobic nature of biochar (*Wang & Wang, 2019*), conducive to the sorption of hydrophobic organic contaminants.

For negatively charged oxyanions such as $NO_3^-$, $PO_4^{3-}$, and $AsO_4^{3-}$, positively charged metal oxides embedding into the biochar facilitates this type of sorption. Moreover, metal oxides increase the active sites in biochar, which is related to the catalytic action of the composite material. Incorporated nitrogen by amino-rich agents also acts as active sites and is linked with such catalytic ability (*Duan, Sun & Wang, 2018*). Especially, transition metal salts increase the magnetism of biochar to meet the separation needs; biochar combining with biofilm is applicable for degradation of some toxic organic contaminants to reduce the toxicity.

## ENVIRONMENTAL CONCERNS AND FUTURE DIRECTIONS

Critically speaking, biochar is not yet widely applied and still in the test stage of researching. At present, the production and application of biochar are not all-pervading, especially in some developing countries where the complete industrial chains are lacking, because of the several environmental concerns that cannot be ignored in the practical application of biochar. In this case, arduous research work needs to be carried out to solve the potential environmental problems and provide the developing countries with exercisable research directions to expand the application of biochar. The potential environmental concerns and propositional future research directions on proposed issues are briefly displayed in Fig. 3.

Although feedstocks for biochar production are extensive and easy to get, these raw materials need to be prepared (grinding, cleaning, and drying) and then pyrolyzed for the available biochar. Modification steps are also required for an ideal sorption effect. Compared with conventional activated carbon, these treatments for biochar will inevitably increase the production cost. Therefore, future researches should attempt to find a compromise between optimizing the production process and maximizing the applicability of biochar to minimize the cost (*Sizmur et al., 2017*). Meanwhile, careful selections of feedstocks, production conditions, and modification methods are critical to acquiring biochars with better performance. The accumulation of a vast quantity of existing research results can help seek the best solutions. For example, the micropore area

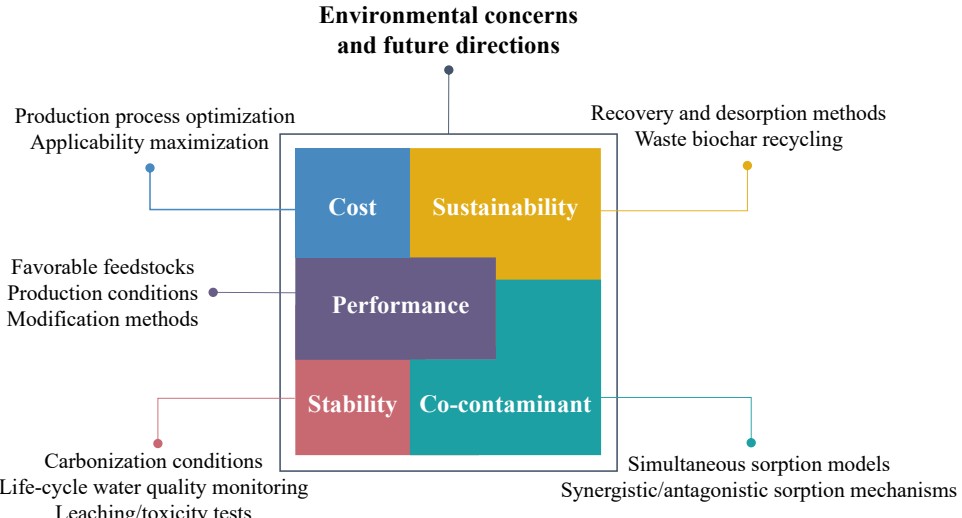

Environmental concerns
and future directions

Production process optimization
Applicability maximization

Cost    Sustainability

Recovery and desorption methods
Waste biochar recycling

Favorable feedstocks
Production conditions
Modification methods

Performance

Stability    Co-contaminant

Carbonization conditions
Life-cycle water quality monitoring
Leaching/toxicity tests

Simultaneous sorption models
Synergistic/antagonistic sorption mechanisms

**Figure 3** **Environmental concerns and future research directions of biochar application.**

of cellulose biochar (280 m$^2$ g$^{-1}$) was larger than that of the lignin biochar (200 m$^2$ g$^{-1}$) when carbonized at the same temperature because of the resistance of lignin, showing the cellulose biomass to be preferable than lignin biomass for biochar production; the surface area and total pore volume of pinewood biochar pyrolyzed at higher temperature were much higher due of the more complete carbonization of lignin, implying that higher temperatures induce well-developed pore structure (*Li et al., 2014*).

The stability of biochar and biochar-based composites should be considered in the practical application of biochar. *Huang et al. (2019)* found the possible dissolution of organic matters from biochar during the complexation with heavy metals, which may increase the carbon content in the water due to the high aromaticity and stability of organic matters. Moreover, the biochars, especially those derived from sewage sludge, could contain high heavy metals that could leach out during the application, causing additional heavy metals contamination (*Wang & Wang, 2019*). For the biochar-based composites, there is a possibility that some of the embedded materials would leach out from the biochar matrix if they are not well-fixed. Considering that the biochar stability generally refers to the stability of its carbon structure (*Wang & Wang, 2019*), studies on the impacts of carbonization conditions on the carbon content and structure need to be conducted. For example, biochar produced via hydrothermal carbonization exhibits higher carbon content than that via gasification and pyrolysis (*Funke & Ziegler, 2010*). Besides, constant water quality monitoring is strongly suggested during the life-cycle application process of the sorbents. Leaching or toxicity tests are proposed using water fleas, alga, fish, or luminous bacteria (*Wang & Wang, 2019*) to determine whether toxic components are dissolving from the biochar.

So far, most researches have focused on the sorption of single contamination in aqueous solutions. However, the prevailing situation in real water application is the

coexistence of a variety of contaminants, where synergistic and antagonistic sorption effects can be observed. The presence of multiple contaminants potentially results in ionic interference and competition of sorption sites, eventually reducing the removal efficiency. At present, empirical data based on sorption of co-contaminant is limited, appealing for the establishment of simultaneous sorption models, which could reveal the involved synergistic or antagonistic sorption mechanisms. To facilitate such studies, reports on biochar sorption should contain sufficient information about the sorbent properties and sorption conditions as detailed as possible in case of providing future directions. Several efforts have been reported, such as the simulated molecular equations for studying competitive sorption of co-contaminant (*Bahamon et al., 2017*); new analysis methods of meta-analysis (*Wang et al., 2019*) and in-depth analysis (*Tran et al., 2019*; *Feng et al., 2016*) have been carried out to develop possible new sorption models.

Although it is consensus that biochar is low-cost, renewable, and sustainable compared with activated carbon (*Mohan et al., 2011*), to achieve the sustainability it is necessary to seek solutions to recovery and desorption of the waste biochar, such as magnetization of biochar, which makes it accessible to separate the contaminant-loaded biochar from water by applying an external magnetic field. However, the desorption of waste biochar may cost a lot. On the other hand, if contaminants sorbed on biochar cannot be effectively desorbed and recovered, it is also feasible to use the waste biochar as a resource, which realizes the recycling of waste biochar in another way. For example, biochar laden with N and P can be of potential use as a slow-release fertilizer in agriculture or ecological remediation (*Roy, 2017*). Accordingly, biochar laden with Cu or Zn can be used as a micro-nutrient fertilizer as well. Nevertheless, attention should be paid whether any harmful components could release from the biochar, which could be sorbed by crops and consequently enter the food chain. Therefore, the safety of applying waste biochar into soil requires further evaluation.

## CONCLUSIONS

This review gives a systematical overview of the broad application of biochar in water and wastewater treatment to remove common and emerging organic and inorganic contaminants. The involved sorption mechanisms are demonstrated as a foundation of studies on biochar sorption behavior. Based on the mechanisms, attention has been paid on biochar modification to improve its performance, which aims to increase the surface area, porosity, or surface sorption sites of the biochar. Exciting frontiers of magnetic biochar and biochar-biofilm combination are also presented. Meanwhile, existing environmental concerns of biochar application are discussed in the aspects of cost, performance, stability, co-contaminant, and sustainability. Finally, future research directions are put forward to facilitate the practical application of biochar.

### Funding

This work was supported by the National Science Fund for Distinguished Young Scholars of China (No. 51925803), the National Natural Science Foundation of China (No. 51878388), Shandong Provincial Key Research and Development Program (Major Scientific and Technological Innovation Project) (No. 2019JZZY010411), the Natural Science Foundation of Shandong Province (No. ZR2018QEE006), and Future Plan for Young Scholar of Shandong University. The funders had no role in study design, data collection and analysis, decision to publish, or preparation of the manuscript.

### Grant Disclosures

The following grant information was disclosed by the authors:
National Science Fund for Distinguished Young Scholars of China:  51925803.
The National Natural Science Foundation of China: 51878388.
Shandong Provincial Key Research and Development Program (Major Scientific and Technological Innovation Project): 2019JZZY010411.
The Natural Science Foundation of Shandong Province: ZR2018QEE006.
Future Plan for Young Scholar of Shandong University.

### Competing Interests

The authors declare there are no competing interests.

### Author Contributions

- Xiaoqing Wang conceived and designed the experiments, performed the experiments, analyzed the data, prepared figures and/or tables, authored or reviewed drafts of the paper, and approved the final draft.
- Zizhang Guo analyzed the data, authored or reviewed drafts of the paper, and approved the final draft.
- Zhen Hu performed the experiments, prepared figures and/or tables, authored or reviewed drafts of the paper, and approved the final draft.
- Jian Zhang conceived and designed the experiments, authored or reviewed drafts of the paper, and approved the final draft.

### Data Availability

   This literature review does not involve raw data or code.

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
