# Peer review of "Recent advances in biochar application for water and wastewater treatment: a review"

_PeerJ, doi:10.7717/peerj.9164_

## Round 0.1 · original submission · Major Revisions

Thank you for your informative submission. I would like to let you know that your manuscript has been recommended for publication subject to major revisions according to the attached referee reports. In addition to our referees, I also would like to emphasize the need for enriching the text with more visual-aids including additional figure(s) and possibly table(s) that can organize the given literature within the text. I also would like to suggest the re-selection of keywords as some of them cover very general topics/terms (i.e. “application”, “modification” and etc). Please note that one of our referees attached an annotated manuscript to their review for your consideration. Once again, thank you for submitting your manuscript to PeerJ and I look forward to receiving your revision.

Reviewer 1 ·

Basic reporting

no comment

Experimental design

no comment

Validity of the findings

The Conclusions section should include the main aspects of biochar modification.

Additional comments

This manuscript reviews the recent advances on biochar applications in water and wastewater treatment. A brief discussion is also conducted on the sorption mechanisms of mainly certain organic pollutants. Finally, a brief discussion on various biochar modification and their effect on the structure, surface properties, and sorption efficiency is presented. The manuscript is well written and scientifically well organized. It is potentially interesting for the readers of the Journal and could be considered for publication after changes.

Specific Comments

1. Since a number of referenced papers concern the use of biochar for the removal of contaminants form water or aqueous solutions the title of the article should be changed to: “Recent advances in the application of biochar for water and wastewater treatment: A review”
2. The effect of pyrolysis temperature on biochar surface and sorption capacity on metals and PAHs as well basic modification of biochar should be benefited by referring to relevant literature on of this topic such as:
a. Preparation and Characterization of Biochar Sorbents Produced from Malt Spent Rootlets." Industrial & Engineering Chemistry Research, 54(39), 9577-9584.2015.
b. Effect of Chloride and Nitrate Salts Effect on Hg(II) Sorption by Raw and Pyrolyzed Malt Spent Rootlets. Journal of Chemical Technology and Biotechnology, 92(8), 1912-1918. 2017.
c. Phenanthrene Removal from Aqueous Solutions Using Well-Characterized, Raw, Chemically Treated, and Charred Malt Spent Rootlets. Journal of Environmental Management, 128, 252-258. 2013.
3. In the section of inorganic ions a brief discussion on the mechanisms for nitrogen and phosphorus removal should be useful for the reader.
4. Line 463. Please clarify what watershed treatment means ?
5. Lines 469-472. Please explain how the addition of biochar into soil reduces runnof and increases runnof time.
6. Modification of biochar. The authors discuss in detail the various methods for biochar modification. In my opinion a comparative discussion is needed at the end of the section on how to select the appropriate modification method according to the target contaminant(s). Which are the future directions for biochar modification?
7. The Conclusions section should include the main aspects of biochar modification.

·

Basic reporting

The manuscript is well-written with minor editing mistakes but the use of the terminology related to sorption can be further improved. Please see suggestions in the annottated pdf file.
Sufficient references are used but some key references are missing and suggestions are provided in the annottated pdf file.
The article is well-structured, the figures are useful and another figure is proposed to organize the last section that is on concerns and future suggestions.

Experimental design

Manuscript content is within the Aims and Scope of the journal.
Rigorous investigation was performed but still some key references are missing. See annottated pdf file.
The structure of the review is well-prepared however some points need to be better highlighed and should be on new separate subsections. See annottated pdf file.

Validity of the findings

The information is well-presented and well-discussed with future concerns and suggestions provided at the end of the manuscript.
The current use of biochar or similar material in wastewater facilities is not sufficinetly -described.

Additional comments

The use of the sorption terminology requires attention and should be carefully used throughout the manuscript.

---

## Round 0.2 · accepted · Accept

Thank you for your re-submission and your great efforts to fulfill our referees’ and my comments. As also further suggested by our referees, I am pleased to inform you that your review manuscript, entitled as "Recent advances in biochar application for water and wastewater treatment: A review” has been accepted for publication in PeerJ. Thank you for your contribution, and we look forward to seeing more of your work in the future. Best wishes.
Ela

Reviewer 1 ·

Basic reporting

The manuscript is well-written.

Experimental design

No comment.

Validity of the findings

No comment

Additional comments

The authors responded very well to reviewer comments.